# Adverse Events in NMOSD Therapy

**DOI:** 10.3390/ijms23084154

**Published:** 2022-04-09

**Authors:** Katrin Giglhuber, Achim Berthele

**Affiliations:** Department of Neurology, School of Medicine, Technical University Munich, Klinikum Rechts der Isar, Ismaninger Str. 22, 81675 Muenchen, Germany; katrin.giglhuber@tum.de

**Keywords:** neuromyelitis optica spectrum diseases, immunosuppressants, rituximab, tocilizumab, eculizumab, inebilizumab, satralizumab, safety profile, adverse events

## Abstract

Neuromyelitis optica spectrum disorders (NMOSD) are rare neurologic autoimmune diseases that have a poor prognosis if left untreated. For many years, generic oral immunosuppressants and repurposed monoclonal antibodies that target the interleukin-6 pathway or B cells were the mainstays of drug treatment. Recently, these drug treatments have been complemented by new biologics developed and approved specifically for NMOSD. In principle, all of these drugs are effective, but treatment recommendations that take this into account are still pending. Instead, the choice of a drug may depend on other criteria such as drug safety or tolerability. In this review, we summarise current knowledge on the adverse effects of azathioprine, mycophenolate mofetil, rituximab, tocilizumab, eculizumab, satralizumab, and inebilizumab in NMOSD. Infections, cytopenias, and infusion-related reactions are most common, but the data are as heterogeneous as the manifestations are diverse. Nevertheless, knowledge of safety issues may facilitate treatment choices for individual patients.

## 1. Introduction

Neuromyelitis optica spectrum disorders (NMOSD) are rare chronic inflammatory diseases of the central nervous system (CNS) that, in recent years, were distinguished from the much more common disease multiple sclerosis (MS) [1]. In the classic form of “neuromyelitis optica” (NMO), the disease is characterised by myelitis or optic neuritis that occur sequentially or coincide. In 2004, an autoantibody against the water channel aquaporin 4 (AQP4) was discovered as a cause of NMO and can be detected in the majority of sufferers [2,3], but possibly not all. This autoantibody is diagnostic and has also shown that several other regions of the CNS may also be affected. Antibodies to AQP4, therefore, define a spectrum of possible disease manifestations, merged under the term NMOSD. Additionally, patients with autoantibodies against MOG (myelin oligodendrocyte glycoprotein) may phenotypically resemble AQP4-antibody-positive NMOSD but differ in course and response to immunomodulatory therapies. Therefore, neuromyelitis optica with evidence of MOG antibodies is currently regarded as another, but separate, entity belonging to the MOG-antibody associated diseases (MOGAD) [4].

Data on the epidemiology of NMOSD are still scarce. Worldwide, women prevail over men, but the prevalence varies widely by geographic region and ethnicity. However, with a prevalence of approximately 0.5 to 10 per 100,000 individuals [1], NMOSD is a rare disease according to the WHO’s definition.

NMOSD has a relapsing course, and recovery from relapses is often incomplete [3]. Because patients often develop significant neurological disabilities early on, the disease can be devastating. Therefore, preventive treatment is a significant medical need.

IgG antibodies against AQP4 play a central role in the pathogenesis of NMOSD [5]. They bind to AQP4 expressed on astrocytes in the CNS, activate the complement system, and trigger the formation of the membrane attack complex, eventually leading to astrocytic and neuronal damage. This inflammatory cascade provides numerous targets for specific immunotherapies; however, these leave the formation of AQP4 in the peripheral immune compartment untouched. And still, the mere presence of AQP4 antibodies does not suffice to trigger NMOSD relapses. Nonspecific immunotherapies, therefore, also have their place in the therapy of NMOSD.

## 2. Neuromyelitis Optica Spectrum Disease: Diagnosis and Treatment

In 2015, an international panel of experts agreed on diagnostic criteria for NMOSD, which not only relied on antibody status but also on a set of core clinical symptoms and magnetic resonance imaging (MRI) findings [6]. The diagnosis of antibody negative NMOSD has remained possible. Especially from a therapeutic point of view, this leads to three categories: AQP4-antibody-positive NMOSD, AQP4-antibody-negative NMOSD, and NMOSD attributable to other disease entities (e.g., MOGAD, Sjögren’s syndrome, or other rheumatic diseases).

For many years, drug therapy for NMOSD was mainly empirical using classical immunosuppressants or therapeutic antibodies targeting CD20 or IL-6. Randomised controlled trials (RCTs) on the efficacy of these drugs were not available, and therefore off-label use has occurred. Corticosteroids, azathioprine, mycophenolate mofetil, and rituximab have been recommended as first-line therapies with reasonable success [1]. In the search for more effective and targeted treatment options for NMOSD and thanks to the orphan drug status, three therapeutic antibodies with differing targets, eculizumab, satralizumab, and inebilizumab have been investigated and approved in recent years [7]. In 2019, the positive phase 3 study of the terminal complement inhibitor eculizumab was published, which was subsequently approved as the first disease-modifying drug for AQP4-antibody-positive NMOSD worldwide. Successful phase 3 trials on satralizumab and inebilizumab followed soon after. Satralizumab was approved by the FDA in 2020 and by the European Commission in 2021 for the treatment of AQP4-antibody-positive NMOSD; inebilizumab was approved by the FDA in 2020 and by the European Commission in 2022 for the same indication.

Now that these new drugs are fully available and all three are unequivocally efficacious, treatment recommendations should ideally rely on comparative data. Unfortunately, concerning treatment efficacy, these data are not available, in the sense that no head-to-head trials have been conducted. Instead, the individual risk of treatment may come to the forefront. To support this, in this review, we aim to provide an overview of the adverse effects of drugs commonly used in NMOSD maintenance therapy (see Table 1). Drugs used to treat acute relapses are not addressed.

Data were gathered from several sources. Firstly, we searched PubMed for the respective drugs in conjunction with neuromyelitis optica or NMOSD as title keywords in any type of publications of the last five years. Secondly, we analysed the FDA approved drug information leaflets and consulted clinicaltrials.com for any recent safety notifications. In addition, for adverse events of special interest (e.g., PML), separate PubMed search queries were performed.

## 3. Classical Immunosuppressive Drugs

### 3.1. Azathioprine

Azathioprine is a prodrug of 6-mercaptopurine and an immunosuppressive antimetabolite. There are single centre reports on azathioprine in NMOSD with up to 103 patients and two meta-analyses covering 977 and 1016 patients [8,9,10,11,12]. Higher case numbers on adverse events (AEs) have been derived from other indications, in particular, inflammatory bowel disease, rheumatoid arthritis (RA), and the early days of kidney transplantation [13,14,15].

According to the official product information approved by the FDA, the frequency and severity of AEs strongly depend on dosing and duration of the treatment, concomitant therapies, and the underlying morbidity [16].

The most common side effects of azathioprine are hematologic and gastrointestinal. Leukopenia or thrombocytopenia can occur at any point during therapy. In NMOSD, leukopenia is reported in around 13% of patients receiving azathioprine [12]. The FDA product information on Imuran^®^ states numbers of >50% for renal homograft patients and 28% of patients with RA [16]. Dose reduction or temporary withdrawal may reverse these effects. However, infections may occur as secondary side effects. In addition, severe leukopenia <2500 cells/mm^3^, anaemia, and thrombocytopenia have been reported [8,9,10]. Patients with deficiencies of the thiopurine S-methyl transferase (TPMT) or the nucleotide diphosphatase (NUDT15), which are both essential enzymes within the inactivation route of azathioprine, are at special risk of severe myelotoxicity [16]. Regarding gastrointestinal side effects, nausea and vomiting frequently occur within the first months of treatment, the percentage in NMOSD patients ranges from 0.8% to 19% [8,9,10,12]. In some cases, these symptoms come with diarrhoea, fever, or myalgias. Administering the drug after meals or dividing drug doses may reduce gastric irritations. Furthermore, up to 23.5% of NMOSD patients treated with azathioprine develop elevated serum alkaline phosphatase, bilirubin, and/or serum transaminases as signs of hepatotoxicity [10]. Other side effects of low frequency include skin rashes, alopecia, arthralgia, steatorrhea, negative nitrogen balance, reversible interstitial pneumonitis, and Sweet’s syndrome [16].

Apart from these AEs, azathioprine also comes with risks of secondary infection and malignancy. However, the descriptions of serious infections in NMOSD patients under azathioprine treatment are rare. Elsone et al. reported three cases of respiratory infections (out of 103 patients, corresponding to 2.9%), Gomes et al. reported on one patient with tuberculosis (out of 19, 5.3%) [9,11]. In inflammatory bowel disease, the reported infection rates range from 2% to 42% [13]. Occasionally, reactivations of latent infections may be severe or even fatal. There are reports of progressive multifocal leukoencephalopathy (PML) by JC virus infection under treatment with azathioprine [17]; however, most of these patients have been treated with other immunosuppressants as well. Notably, one NMOSD patient with PML under monotherapy with azathioprine has been described [18].

Regarding the risk of secondary malignancies, the data suggest that it is elevated under azathioprine therapy, however, much more after renal transplantation than in other diseases [19]. Yet, there is a controversy on the causality, as the numbers vary strongly between morbidities, and autoimmune diseases themselves may contribute to an elevated malignancy risk. Regarding RA, the FDA states that “it has not been possible to define the precise risk of malignancy” in patients treated with azathioprine [16]. Regarding NMOSD, Costanzi et al. reported three lymphoma cases in 99 patients treated with azathioprine [8]. Gomes et al. observed one case of breast cancer (out of 19 patients) [11]. One large review on patients with inflammatory bowel disease found cancer in 7 out of 3931 patients treated with azathioprine, among them two fatal cases with Hodgkin’s and hepatosplenic lymphoma [13].

### 3.2. Mycophenolate Mofetil

Mycophenolate mofetil (MMF) also belongs to the drug class of immunosuppressive antimetabolites. The FDA product information is based on five studies that examined the use of MMF in around 1500 transplant patients to prevent organ rejection. In these, “the most common adverse reactions in clinical trials (20% or greater) include diarrhoea, leukopenia, infection, vomiting, and there is evidence of a higher frequency of certain types of infections, for example, opportunistic infection” [20].

Regarding the use in NMOSD, there has been one meta-analysis of 11 studies that summarised 106 AEs in 594 patients, corresponding to a rate of 17.8% [21]. The largest proportion of data was derived from Huang et al. with 90 patients, of which 43% of the patients developed adverse events [22]. Interestingly, the observed rate varied considerably. One study reported only 3 out of 62 patients with mild AEs, namely mild hair loss and mild liver enzyme elevations [23]. Considering the pooled data of Songwisit, the most common AEs in NMOSD were infections in 33/594 patients [21]. The leading infections were respiratory infections and pneumonia, followed by urinary tract and herpes zoster infections. Huang reported that 3 out of 21 infections were severe, i.e., two patients suffering from pneumonia of unreported aetiology needed ventilation and one patient developed a hemorrhagic varicella infection three months after MMF initiation and died from respiratory distress syndrome [22]. To the best of our knowledge, there were no other cases of serious viral infections such as, for example, JCV-associated PML, cytomegalovirus infections, or viral reactivation with Hepatitis B or C [21,24].

The next most frequent side effects were gastrointestinal, including elevated liver enzymes in 4.5–20% of patients, diarrhoea and other digestive complaints in approximately 2% of patients [21,22,25]. In other indications, gastrointestinal bleeding requiring hospitalisation, ulcerations, and perforations were also observed in clinical trials [20]. Bone marrow suppression was observed in 0.9–2.7% of NMOSD patients treated with MMF and included foremost anaemia and leukopenia [21,22]. The symptoms and laboratory abnormalities were mostly reversible after discontinuation of treatment. Considering quality of life and wellbeing, mild hair loss in around 2% of patients is an issue [21,22]. Interestingly, Huang et al. showed, by means of regression analysis, that adverse events of MMF therapy significantly decreased in combination with glucocorticoids [22].

Like azathioprine, MMF for immunosuppression has been shown to increase the risk of developing malignancies, especially lymphomas or skin cancer [20]. Huang et al. reported one case of rectal cancer discovered by elevated levels of the carcinoembryonic antigen during MMF therapy in a patient with NMOSD [22]. Poupart et al. described one fatal case of lung cancer in an NMOSD patient with a smoking history and six years of MMF treatment [24]. Causalities could not be proven in both cases.

## 4. Rituximab

Rituximab is a monoclonal antibody against CD20. It is approved for the treatment of non-Hodgkin lymphoma, chronic lymphocytic leukemia, RA, granulomatosis with polyangiitis, microscopic polyangiitis, and pemphigus vulgaris. Regarding RA, there are data from clinical studies on >2500 patients. “Very common” adverse events (at least 1/10 of patients treated) were infections of the upper respiratory and the urinary tract, infusion-related reactions (IRR), headache, and reduced serum levels of IgM [26].

Regarding NMOSD, there are several meta-analyses available, which slightly differ regarding the trials selected [27,28,29,30]. The meta-analyses all covered the study of Kim et al. [31], which has been the largest open-label trial with 100 patients included. Additionally, there have been two randomised controlled trials. Nikoo et al. compared rituximab with azathioprine [32] in terms of efficacy, and the RIN-1 study by Tahara et al. compared rituximab with a placebo [33]. Only recently, the Canadian Agency for Drugs and Technologies in Health (CADTH) published a Health Technology Report on rituximab in NMOSD, which covered safety as well [34].

Depending on the depth of data collection, the share of patients with AEs was reported to be 16.46% [29] up to 28.57% [30]. Most common were IRR in 10–13% of patients and infections in around 9% of patients [28,32]. Interestingly, the RIN-1 study observed AEs in 17/19 patients, i.e., in 90% [26]. Most of them were mild and infusion related. From the RA studies, it has been reported that IRR most often occurred after the first administration of rituximab, and premedication with glucocorticoids alleviated frequency and severity [26].

Infections mainly involve upper respiratory tract and urinary tract infections. Additionally, there is a risk of reactivating opportunistic infections such as herpetic rash and tuberculosis [27]. In RA, there have been reports of fatal PML under therapy with rituximab [35]; however, in NMOSD, to the best of our knowledge, there are as yet none. Severe infectious AEs have occurred in 2–3% of patients, among them severe pneumonia, septicemia, and severe allergic reactions [29,32]. Again, Tahara et al. [33] reported much higher percentages with severe adverse reactions in 3/19 patients (16%). However, apart from one adverse reaction (an infection of the foot not further classified), the listed severe events (lumbar compression fracture, diplopia, and uterine cancer) may count among very rare or even unrelated side effects [33].

Interestingly, there are data on treatment-related mortality. Mortality was reported in one review with 1.6% without further clarification of the causes of death [28]. A second review reported that 5/577 (0.9%) of NMOSD patients treated with rituximab died, of whom two patients suffered from pneumonia, one patient suffered from a urogenital infection and thrombosis, one patient died after bone marrow transplantation, and one patient died of cardiac and respiratory failure due to severe myelitis with rostral extension [29]. A third review reported one death by septicemia and one death after cardiovascular failure [27,36,37]. Touching upon the latter, for RA, severe cardiovascular events were reported to be equally frequent in patients receiving rituximab or a placebo [26].

Further generic but important side effects of rituximab treatment include neutropenia, mainly mild to moderate, and low serum titers of IgG and IgM, at least over time. Hypogammaglobulinemia, however, is still debated as to whether and how it correlates with an increased risk of any or severe infections [26,38].

## 5. Tocilizumab

Tocilizumab is a monoclonal antibody against the interleukin-6 receptor that is approved to treat several rheumatic disorders, including RA. The FDA approved product information on tocilizumab makes use of data derived from five studies with more than 4000 patients [39]. Among these, only 288 patients received tocilizumab as a monotherapy, whereas the majority was on a combination with, for example, methotrexate. The most-reported AEs were upper respiratory tract infections, nasopharyngitis, headache, hypertension, and increased transaminase levels. Serious AEs primarily comprised infections, including pneumonia, urinary tract infections, cellulitis, herpes zoster, gastroenteritis, diverticulitis, sepsis, and bacterial arthritis. Moreover, the product information points to opportunistic infections such as tuberculosis, cryptococcosis, aspergillosis, candidiasis, and pneumocystosis which may occur.

Regarding the safety of tocilizumab in NMOSD, there are several case reports and a few case series. However, data are often blended by the concomitant use of other immunosuppressives such as azathioprine or MMF, and the diligence of safety reporting varies [40,41,42,43]. In 2020, Zhang et al. published their results on the TANGO study, the first and only randomised controlled trial on tocilizumab versus azathioprine for the treatment of NMOSD [44]. Two systematic reviews have summarised the available data in NMOSD [45,46]. Lotan et al. described that 11 out of 17 studies reported no AEs. In the remaining studies, leukopenia, elevated cholesterol levels, urinary tract infections, and anaemia were the foremost AEs [45]. Xie et al. described 75 AEs in a total of 89 patients, including the TANGO trial (84%), which contributed 36 treatment-related AEs in 59 patients (61%) [46]. The AEs were mostly mild: 28–29% of patients developed urinary tract infections, 21–29% of patients developed upper respiratory tract infections, 24–31% of patients developed mild liver enzyme elevations, and 6–10% of patients developed elevated cholesterol levels or leuko-/lymphopenia [44,46].

Of note, in a case series of 57 NMOSD or MOGAD patients treated with tocilizumab for a median of two years, pneumonia was more frequent in patients with add-on immunosuppressants (18% versus 6% of patients under tocilizumab monotherapy). Four patients suffered a flare-up of concomitant autoimmune diseases [43].

In the TANGO trial, five SAEs occurred (in 59 patients = 8%), including severe pneumonia, herpes zoster, deep vein thrombosis, cerebral haemorrhage, and myelitis, none with a fatal outcome. There were two deaths reported, one each under tocilizumab and azathioprine. The two deaths were both classified as not related to treatment, i.e., one as due to “central respiratory failure secondary to myelitis involving the high cervical spine and medulla oblongata”, the other to “severe myelitis” [41,44].

As for secondary malignancies, the FDA product information generally advises that immunosuppressives may increase the risk. However, the specific impact of treatment with tocilizumab is unknown [39], and no cases in tocilizumab-treated NMOSD patients have been reported yet.

Finally, hypersensitivity reactions must be mentioned as a potential risk. Anaphylaxis and other hypersensitivity reactions that required treatment discontinuation were reported in <1.0% for RA, among them fatal cases [39]. For NMOSD and MOGAD, Ringelstein et al. reported infusion-related reactions (headache, abdominal pain, as well as vertigo/nausea) in 7/57 patients treated with tocilizumab [43].

## 6. Eculizumab

Eculizumab is an IV monoclonal antibody against the complement protein C5, and FDA-approved, since 2019, for treating AQP4-antibody-positive NMOSD. Safety data are available from a small pilot study, the placebo-controlled pivotal trial PREVENT, and its open-label extension (OLE) study [47,48,49].

Summarised data from the latter two studies came from the treatment of 137 AQP4-antibody-positive NMOSD patients for a total of 362.3 patient years (PY) until June 2020 [49]. The rates of treatment-related AEs were similar in the eculizumab group of the PREVENT trial and during the OLE; altogether, there were 183.5 AEs in 100 PY (as compared with 167.5 AEs in 100 PY in the PREVENT placebo arm). Most common AEs were headache (57.7 events in 100 PY), nasopharyngitis (27.6 in 100 PY), upper respiratory tract and urinary tract infection (25.7 each in 100 PY), back pain and diarrhoea (12.4 each in 100 PY), nausea (13.2 in 100 PY), and arthralgia (8.8 in 100 PY). The rate of treatment-related SAEs was 8.0 in 100 PY. SAEs included severe infections, which occurred in 25/137 patients, corresponding to 10.2 events in 100 PY. Among these, five patients suffered from pneumonia, four patients each from urinary tract infection or cholecystitis, and two patients each from cellulitis or sepsis [49].

Interestingly, a subgroup analyses of the PREVENT double-blind phase revealed that serious infections occurred less frequently with eculizumab than with a placebo, regardless of concomitant immunosuppressive medications or prior rituximab use [50]. There was one reported death in the PREVENT trial; the patient died under medication with eculizumab and azathioprine due to pleural empyema [47]. The event was categorised as possibly related to eculizumab treatment. During the OLE, no fatalities were reported.

Eculizumab is known to especially increase the risk of infections with encapsulated bacteria such as meningococci. The risk has been estimated to be 2000-fold higher than in the untreated population [51]. There have been reports of meningococcal infections despite proper vaccination, hereunder one NMOSD patient in the pilot study with eculizumab [48,52]. Neither the PREVENT study nor the OLE observed meningococcal infections. However, one patient in the OLE developed an infection due to Neisseria gonorrhoeae [49].

Furthermore, the FDA product information of Soliris^®^ warns of infections by Aspergillus species [51]. These are reported in patients with atypical hemolytic uremic syndrome treated with eculizumab, including one fatality due to a ruptured cerebral aneurysm related to Aspergillus infection [53].

Being a therapeutic protein, eculizumab may be prone to trigger infusion-related reactions. Notably, the rate of IRR is rather low, and symptoms are mild and transient. In clinical trials, no patient experienced an infusion-related reaction requiring eculizumab discontinuation [47,51].

## 7. Satralizumab

Satralizumab is a monoclonal antibody against the IL-6 receptor, which was specifically developed for the treatment of NMOSD and approved by the FDA in 2020.

It is administered subcutaneously and, as a so-called “recycling-antibody”, is superior to tocilizumab in terms of pharmacokinetics. The drug proved its efficacy in two international phase 3 RCTs: “SAkuraStar” with satralizumab as monotherapy and “SAkuraSky” with Satralizumab as an add-on to a stable immunosuppressant treatment (azathioprine, MMF, oral corticosteroids) [54,55]. These studies were both designed to have an OLE period, and safety data including this extension until February 2021 is already available; it refers to altogether 166 patients treated with satralizumab for 437.7 PY [56,57]. Safety data pooled from both double-blind and OLE periods contain 418.8 AEs per 100 PY; most frequent AEs were all upper respiratory tract infections (25.1 in 100 PY), nasopharyngitis (20.1 in 100 PY), urinary tract infection (18.5 in 100 PY), and headache (11.0 in 100 PY) [58].

Similarly, the FDA product information lists the following AEs observed in 15% or more of patients: nasopharyngitis, headache, upper respiratory tract infection, gastritis, rash, arthralgia, extremity pain, fatigue, and nausea [59].

Injection-related reactions occurred with 12.1 events per 100 PY, predominantly of mild to moderate severity. In SAkuraSky, injection-related reactions were more frequent in the satralizumab than in the placebo group, although treatment discontinuation was not required [55]. 

The rate of infections and serious infections did not differ considerably between patients treated with satralizumab and those on a placebo. Overall, there were 112.4 infections per 100 PY and 3.9 serious infections per 100 PY [58]. There were no opportunistic infections in SAkuraStar [54]. In SAkuraSky, herpes zoster infections were observed at similar rates for satralizumab and a placebo, and all infections were mild or moderate [55]. By looking at the pooled data of both RCTs, Greenberg et al. concluded that there was no increased risk of opportunistic infections [56]. Furthermore, the OLE periods, with a follow-up to year 6, did not show an increased risk of infections or serious infections over time [57].

Severe AEs were reported with 15.1 events per 100 PY in the pooled dataset [48]. In SAkuraStar, severe AEs were reported at a higher rate for satralizumab than a placebo; however, always system organ classes affected were widespread, 73% of events were unrelated to treatment, and no patient discontinued treatment due to the SAE [54]. There were no cases of PML, no deaths, and no anaphylactic reactions.

The FDA product information on Enspryng^®^ additionally warns of laboratory abnormalities including neutropenia, thrombocytopenia, liver enzyme and lipid level elevations, and fibrinogen and complement deficits [59]. However, the pooled satralizumab data indicate that these abnormalities were mainly transient or intermittent. Furthermore, grade 3 or 4 neutrophil count decreases were not associated with serious infections, thrombopenia or fibrinogen decrease was not associated with bleeding events, and liver function testing did not indicate significant drug-induced liver injury [59].

## 8. Inebilizumab

Inebilizumab is the first-in-human monoclonal antibody targeting CD19. Given intravenously, it depletes the B-cell lineage more profound than rituximab, for example, including plasma cells. The FDA approval for the treatment of NMOSD in 2020 is based on the pivotal phase 3 trial N-MOmentum [60]. It entailed a randomised, placebo-controlled period with 174 NMOSD patients receiving inebilizumab for 6.5 months and an OLE with altogether 213 patients [60]. The FDA’s product information refers to data from 324 PY of 208 patients [61].

During the placebo-controlled phase of N-MOmentum, 125/174 patients had an adverse event [60]. Across the core phase and OLE, AEs with an incidence >10% included: urinary tract infections (20%), nasopharyngitis (13%), infusion reactions (12%), arthralgia (11%), and headache (10%) [61]. All infusion reactions were mild or moderate, and they were most common with the first infusion [60]. In the core study, 8/174 patients had 10 SAEs, namely arthralgia, atypical pneumonia, third-degree burns, acute cholangitis, acute cholecystitis, diarrhoea, abnormal hepatic function, myelitis, urinary tract infection, and blurred vision. In follow-ups, there were no new safety concerns after treating with inebilizumab for an average of 2 years (range of 0.2–4.4) [62] and 4.5 years (75 AQP antibody-positive patients, range of 4.01–5.53) [54], and no AE led to the discontinuation of treatment [63].

However, after no deaths in the placebo-controlled phase of N-Momentum, two patients died during the OLE. One former placebo patient suffered severe pneumonia followed by an NMOSD attack, then entered the OLE, received the first dose of inebilizumab, but died nine days later, most probably due to “respiratory insufficiency caused by the recent NMOSD attack”. The other patient received inebilizumab from the beginning, had an adjudicated relapse, entered the OLE, developed new neurological symptoms including aphasia and seizures, had a respiratory arrest, and died of cardiopulmonary complications. Magnetic resonance imaging showed new large cortical and subcortical lesions; brain biopsy was not done. Cerebrospinal fluid was twice negative for JC virus in certified external laboratories, once positive in a local laboratory. The diagnosis remained open, as the question of whether inebilizumab was causative or not [60]. There are no other cases of suspected or confirmed PML under therapy with inebilizumab [61].

As is the case with other B-cell-depleting antibodies, the risk of infections is increased by inebilizumab. Still, opportunistic infections have not been reported, nor were reactivations of viral infections. Yet, chronic hepatitis was one of the exclusion criteria of the N-MOmentum trial [61]. There are some additional warnings on laboratory abnormalities. First, inebilizumab reduces the lymphocyte counts, as per the mode of action. At the end of the core phase of the N-Momentum trial, 5.3% of patients treated with inebilizumab had a lymphocyte count below the lower limit of normal (LLN) (as compared with 4.2% of patients receiving a placebo) [61]. Furthermore, neutrophil counts may decrease: 6.9% of patients developed mild neutropenia (absolute neutrophil count 1.0–1.5 *×* 10^9^/L) and 1.9% moderate neutropenia (0.5–1.0 *×* 10^9^/L). At the end of the placebo-controlled phase, 12% of inebilizumab-treated patients had a neutrophil count below LLN as compared with 4.2% of patients in the placebo arm [61]. The reductions were generally transient and not associated with serious infections [60].

Finally, inebilizumab is expected to lower immunoglobulin levels. Notably, this could be seen quite early in patients treated with inebilizumab as compared with other B-cell depleting therapies, where this is known as a side effect with a relatively late onset. With inebilizumab, at the end of a 6.5-month core study, the total immunoglobulin level was reduced by about 8% from baseline, IgG was reduced by 4%, and IgM was reduced by 32%. In the combined core study and OLE safety analysis, IgG levels below the limit of normal were observed in 6.6% of patients after one year and in 13% of patients after two years, and for IgM, in 31% and 42% of patients, respectively [61]. IgG levels continued to decrease steadily over a 4.5 year follow-up; however, it is not yet known whether and when hypogammaglobulinemia also leads to an increased rate of infections [63].

**Table 1 ijms-23-04154-t001:** Adverse events of high frequency according to the FDA- and EMA-approved prescribing information leaflets. Taking into consideration a dissimilar use of the label “high frequency”, percentages are specified, respectively. * Data from patients with rheumatoid arthritis.

Drug	Adverse Events of High Frequency				
	Infections	Gastrointestinal Side Effects	Leukopenia	Arthralgia	Other
**Azathioprine *** [16]	-	≥10%	≥10%	-	-
**MMF** [20]	≥20%	≥20%	≥20%	-	-
**Rituximab *** [26]	≥10%	-	-	-	-
**Tocilizumab** [39]	≥5%	-	-	-	Headache, hypertension, increased liver enzymes, injection site reactions (≥5%)
**Eculizumab** [51]	≥10%	≥10%	-	≥10%	Back pain, contusion, dizziness (≥10%)
**Satralizumab** [59]	≥15%	≥15%	-	≥15%	Extremity pain, fatigue, headache, rash (≥15%)
**Inebilizumab** [61]	≥10%	-	-	≥10%	-

## 9. Comparative Data on Safety

Regarding efficacy, several publications have attempted to compensate for the lack of proper head-to-head trials by using meta-analytic methods, for example, [64,65,66].

Unfortunately, concerning safety, there are only a few studies in which data from different substances have been compared side-by-side. Giovannelli et al., in their recent meta-analysis on the efficacy of immunosuppressants in NMOSD, decided against analysing safety data because the data sources were regarded to be too heterogeneous [67]. Huang et al. managed to feed AE data from azathioprine, mycophenolate mofetil, rituximab, and cyclophosphamide into a network meta-analysis. MMF was superior to rituximab, which was superior to azathioprine in terms of adverse events (as a measure of tolerability) [68]. Finally, Kong et al. performed a classical meta-analysis of the risk of AEs for the monoclonal antibodies. Inebilizumab, eculizumab, and rituximab showed a trend in favour of the treatment as compared with a placebo; tolicizumab showed a trend in favour of treatment compared to azathioprine; and the data from the two RCTs on satralizumab were contradictory [69].

## 10. Drug Interactions

Drug interactions may limit the use of immunomodulatory drugs and bring the patient’s comorbidities into play. For this, coexisting autoimmune diseases in NMOSD may become operational. Additionally, immunosuppressants or monoclonals are often combined with each other or, at least temporarily, with steroids. This may add or even potentiate the risk of side effects, foremost the risk of infections.

Azathioprine is a prodrug that is almost completely cleaved in its primary active metabolite 6-mercaptopurine (6-MP); 6-MP is inactivated via methylation catalysed by the enzyme thiopurine S methyltransferase (TPMT) or oxidation by the enzyme xanthine oxidase (XO). The latter pathway would be inhibited by the concomitant use of xanthine oxidase inhibitors, the most prominent of which is allopurinol. Thus, this combination should be avoided whenever possible to prevent toxic plasma concentrations of azathioprine or 6-MP. In patients who need to have both drugs, the dosage of azathioprine must be reduced to one third or one quarter, and signs of toxicity have to be monitored closely. Moreover, aminosalicylates (e.g., sulphasalazine, mesalazine, or olsalazine) may inhibit the TPMT leading to the accumulation of azathioprine and metabolites [16].

Mycophenolate mofetil is readily absorbed and hydrolysed to the active metabolite mycophenolic acid (MPA). Drug interaction studies have shown that magnesium- or aluminium hydroxide-based antacids decreased the absorption of MMF, and the use of proton pump inhibitors decreased plasma levels of MPA. Cyclosporine may as well lower MPA concentrations. Drugs that are excreted renally might compete with the renal elimination of MMF metabolites. For example, when used with acyclovir, ganciclovir or valacyclovir, signs of either toxicity may occur. Further, drugs that hamper the enterohepatic recirculation may interfere (e.g., antibiotics). Finally, MMF might reduce the action of oral contraceptives [20].

Direct drug interactions are neither reported for rituximab, inebilizumab, nor eculizumab [26,51,61]. Of interest, targeting the IL-6 pathway with Tocilizumab may indirectly interfere with comedications: Inhibition of IL-6 may upregulate the activity of several isoforms of the CYP450 enzyme system. This may result in decreased levels of, for example, omeprazole, warfarin, statins or cyclosporine or oral contraceptives [39].

## 11. Pregnancy

There are no drugs used in the treatment of NMOSD regarded to be completely safe or proven to be innocuous to the fetus when used in pregnant women (see Table 2). Thus, at least all females of child-bearing potential shall be advised to use adequate contraceptive methods during treatment.

Nevertheless, since NMOSD requires lifelong therapy and since this treatment is generally quite successful in reducing the burden of disease, family planning is, of course, an issue [70]. The tangible pregnancy risk conveyed by the drugs, therefore, have to be weighed individually. Whereas MMF is strictly contraindicated, information on eculizumab and tocilizumab in pregnancy is quite limited, and the data on satralizumab and inebilizumab are simply insufficient to draw any conclusions. In contrast, azathioprine and rituximab, in the hands of collaborating expert neurologists and obstetricians, are regarded as relatively safe. Regarding azathioprine, there is a large body of data gathered from female as well as male patients with inflammatory bowel disease, indicating that azathioprine may slightly increase the odds of preterm delivery but is safe otherwise [71]. For rituximab, a comprehensive case series of 88 pregnancies was recently published and may give initial guidance [72]. However, more data are needed.

## 12. Conclusions

The drug arsenal for the treatment of NMOSD has become much larger in recent years. In addition to the classic immunosuppressants, targeted biologics are now also available. The latter seem to be preferable due to their specific mode of action, but a significant proportion of NMOSD patients could also be stabilised with conventional drugs so far available. Therefore, safety aspects may help set priorities.

Unfortunately, the systematic description of the side effects of immunomodulatory treatment in NMOSD has its limitations. Interestingly, the seven drugs considered in this review show no fundamental differences in tolerability. In all of them, as a downside of the desired immunomodulatory effect, a varying tendency to infections can be observed, and, in the case of the pulsed injectable drugs, infusion/injection reactions. Most reported infections affect the urinary and respiratory tracts. Although not uncommon in principle, NMOSD patients may be predisposed to these infections due to disease-specific symptoms or disabilities such as bladder dysfunction, motor dysfunction, or dysphagia following myelitis. In addition, some symptoms recorded as an adverse effect of drug treatment may just as well be a symptom of the disease itself, such as pain or fatigue. Apparent compound-specific side effects are occasionally known (e.g., hypogammaglobulinemia with B-cell targeting therapies or the risk of infections due to encapsulated pathogens with eculizumab), but they are (still) of unknown significance or only rarely described.

The endeavour to compare the precise frequencies of general side effects between substances turns out to be difficult. The reason for this is, on the one hand, the heterogeneity of the data sources, which is in a way like comparing apples and oranges. On the other hand, for conventional immunosuppressants, safety data come from long-standing registries and drug monitoring systems that are fed independently of the indication for which the drugs were used. Thus, making a statement about tolerability in NMOSD itself is fraught with uncertainty. In addition, these registry data often have a reporting bias, and comparisons to patients who did not receive the respective drug in the same clinical situation are missing. Therefore, the contribution of the underlying disease to the tolerability of the therapy remains open.

For the new biologics that have been explicitly developed and approved for the treatment of NMOSD, the situation is different. Side effects, complications, and tolerability were systematically and prospectively recorded here in the RCTs conducted. This provides a very complete picture of the side effects profile and, most importantly, also allows showing how frequently the respective side effects were reported by patients who did not receive the drug. This comparison can put the risk of side effects into perspective but rarely also lead to self-contradictory findings (e.g., in the satralizumab SAkura studies [69]). Further, a disadvantage of safety data obtained in RCTs is that the patient population treated there was selected and does not correspond to the therapeutic “real world” situation, where the patients are in part significantly more advanced in the disease, possibly older or have other comorbidities. In short, they may have a higher risk of side effects per se.

Finally, the safety data also differ in the availability and extent of long-term data. While safety data for the classical immunosuppressants and B-cell/Il6-targeted antibodies could be collected over many years or decades, the observation period for the new biologics such as eculizumab and, especially, satralizumab and inebilizumab is still very short. There is, therefore, a risk that so far unknown but significant new safety issues are still to be uncovered when the new drugs are used on a larger scale.

In addition, there are almost no data available on an issue that can also be described as a side effect in the broadest sense: a characteristic of NMOSD is that many of the affected patients suffer from other autoimmune diseases concomitantly (e.g., Lupus or Sjogren’s disease). Here, it is largely unknown how often flare-ups of these co-diseases occur during treatment with the individual drugs, i.e., whether drugs may treat these co-diseases inadequately. It is to be expected that this “side effect” would become more frequent the more a particular drug is directed at a target specific to NMOSD. Conventional immunosuppressants may offer advantages in this respect, but there is no robust data on this.

Thus, neither the criterion of efficacy alone nor the criterion of safety alone is sufficient for the selection of a drug for immunomodulatory therapy of NMOSD. The available drugs are all effective, and all drugs have side effects. Instead, a set of criteria need to be considered in the individual selection, as well as in the formulation of treatment recommendations: in addition to any aspects of varying efficacy and tolerability, general and autoimmune comorbidities, the rapidity of drug effects, the mode of action and route of application (tablets or parenteral drugs and continuous versus pulsed therapies), and the logistics or availability (e.g., rural versus urban areas and reimbursement) have to be considered. Finally, this set of criteria has to be reiterated for AQP4-antibody-negative NMOSD patients because the new biologics are not available for them.

Notwithstanding, it is most relevant for clinical practice to be informed about the side effects of the drugs we use and to collect more data on this important topic.

## Figures and Tables

**Table 2 ijms-23-04154-t002:** Advice on the use of drugs in pregnancy, quoted from the FDA-approved prescribing information leaflets.

Drug	Use in Specific Populations: Pregnancy
Azathioprine [16]	Can cause fetal harm when administered to a pregnant woman. Should not be given during pregnancy without careful weighing of risk versus benefit.
Mycophenolate mofetil [20]	Use of MMF during pregnancy is associated with an increased risk of first trimester pregnancy loss and an increased risk of multiple congenital malformations in multiple organ systems.
Rituximab [26]	Can cause fetal harm when administered to a pregnant woman. Rituximab can cause adverse developmental outcomes including B-cell lymphocytopenia in infants exposed to Rituximab in-utero.
Tocilizumab [39]	The limited available data with Tocilizumab in pregnant women are not sufficient to determine whether there is a drug-associated risk for major birth defects and miscarriage. Monoclonal antibodies, such as Tocilizumab, are actively transported across the placenta during the third trimester of pregnancy and may affect immune response in the in utero exposed infant.
Eculizumab [51]	Limited data on outcomes of pregnancies that have occurred following Eculizumab use in pregnant women have not identified a concern for specific adverse developmental outcomes. Animal studies using a mouse analogue of the Eculizumab molecule (murine anti-C5 antibody) showed increased rates of developmental abnormalities and an increased rate of dead and moribund offspring at doses 2–8 times the human dose.
Satralizumab [59]	There are no adequate data on the developmental risk associated with the use of Satralizumab in pregnant women.
Inebilizumab [61]	Inebilizumab is a humanised IgG1 monoclonal antibody and immunoglobulins are known to cross the placental barrier. There are no adequate data on the developmental risk associated with the use of Inebilizumab in pregnant women. However, transient peripheral B-cell depletion and lymphocytopenia have been reported in infants born to mothers exposed to other B-cell depleting antibodies during pregnancy.

## Data Availability

Not applicable.

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
