# Peer review of "Adverse Events in NMOSD Therapy"

_ijms, 2022, doi:10.3390/ijms23084154_

Round 1
Reviewer 1 Report
The authors present a review of adverse events in therapies commonly used in NMOSD. Brief description of the background of the disease and its consequences, as well as recent progress in its treatment, justify significance of the topic and its clinical relevance.
The review includes classical immunosupressants and new generation agents, approved specifically for NMOSD on the basis of clinical trials. The differences between these two categories of therapies were outlined, concerning their mode of action but also sources of data about their effectiveness and safety. Adverse events of all these therapies were listed and extensively discussed. Separate sections covered drug interactions, pregnancy-related issues and diseases coexisting with NMOSD, all of which indeed deserve special attention in this field.
The references are sufficient in number, up-to-date and properly cited. The manuscript is clearly written and well organized.
Overall, the review is very comprehensive and has a high didactic potential. Minor issues need to be addressed and a few amendments seem necessary to enhance its value, which are listed below.
- Methodology of the review should be provided – how were the sources searched and selected?
- It should be mentioned in the introduction that the review is focused only on maintenance treatment of NMOSD ( and does not cover treatment of relapse – corticosteroids, plasma exchange etc)
- Specificity of NMOSD should be more highlighted throughout the text, especially in the following aspects:
With regard to each medication, I would suggest to separate more clearly AE reported from their use in other immune-mediated diseases from those observed in NMOSD (in the text and Table 1). Are these reports consistent or divergent? Are discrepancies due to relatively small numbers of patients with NMOSD and their short-term follow-up or to the features of the disease itself ?
Do the background of the disease (humoral autoimmune response) or its nature (involvement of CNS) predispose for particular AE ? (e.g. bladder dysfunction resulting from myelitis facilitates urinary tract infections)
How serious is the problem of differential diagnosis: NMO relapse vs AE of treatment (e.g. opportunistic infections of CNS)? Myelitis and blurred vision, reported as AE in inebilizumab therapy, make a good point for discussion about such controversies.
- In the final chapter a concept of evaluating benefit –to-risk ratio in therapeutic decisions should be developed. Considering relatively good safety profile, control over devastating disease activity seems a priority of therapeutic strategy. However, the problem of seronegative NMOSD patients, who are less likely to benefit from the new therapies but still may be exposed to AE, deserves more attention.
Reviewer 2 Report
The authors presented a useful review of adverse events in NMOSD therapy.
However, the review needs some corrections before further consideration:
Please insert references inside the tables (1 and 2), not below.
